# Pain Assessment in Cattle by Use of Numerical Rating and Visual Analogue Scales—A Systematic Review and Meta-Analysis

**DOI:** 10.3390/ani14020351

**Published:** 2024-01-22

**Authors:** Theresa Tschoner, Kristina R. Mueller, Yury Zablotski, Melanie Feist

**Affiliations:** 1Clinic for Ruminants with Ambulatory and Herd Health Services at the Centre for Clinical Veterinary Medicine, Ludwig-Maximilians-Universität Munich, Sonnenstrasse 16, 85764 Oberschleißheim, Germany; y.zablotski@med.vetmed.uni-muenchen.de (Y.Z.); melanie.feist@lmu.de (M.F.); 2School of Veterinary Science, Massey University, Private Bag 11 222, Palmerston North 4474, New Zealand; k.mueller@massey.ac.nz

**Keywords:** acute pain, analgesia, calves, dairy cattle, farmers, NRS, pain management, pain scoring, survey, VAS, veterinarians

## Abstract

**Simple Summary:**

Pain assessment in cattle can be performed using pain scales, e.g., the Numerical Rating (NRS) or Visual Analogue Scale (VAS). Pain scoring via pain scales is subjective to the experience and attitude of the observer. This systematic review and meta-analysis aimed to compare pain evaluation by dairy practitioners of different countries participating in surveys about pain management in cattle. Pain scoring is influenced by different factors, including the participant’s age, gender, education, and profession. Female participants gave higher pain scores, as did recently graduated veterinarians. Differences in pain scales, nomenclature of terms, and values used between studies complicate the direct comparison of pain scores. The majority of these articles originated from the European Union. Different legislation and welfare considerations of countries could possibly influence pain scoring. Only a small number of studies could be included in the meta-analysis. Mean values of pain scores given for different procedures and conditions differed significantly, for both Numerical Rating Scales 0–10 and 1–10. The findings of the present study showed that the comparison of pain scores used in different articles is difficult due to the use of different pain scales and nomenclature, and that pain scoring is influenced by different factors, such as age and gender.

**Abstract:**

Subjective pain assessment in cattle is contingent upon the observer’s experience and attitude. Studies of pain assessment in cattle by veterinarians and farmers using different pain scales have been published. This systematic review and meta-analysis aimed to describe and compare the pain scores given by veterinarians and producers for different procedures and conditions using either a NRS or VAS. The literature search was conducted with PubMed (MEDLINE) and Agricola, using defined search terms (e.g., peer-reviewed). A total of 842 articles were identified. After screening of duplicates, abstracts, and full texts, a total of 16 articles were included in this systematic review. Different pain scales were used for the included studies (NRS 0–10 for eight studies, NRS 1–10 for six studies, NRS 1–10 and VAS 0–10 for one study, and VAS 0–1 for one study). Most studies (*n* = 11) originated from the European Union. Mean values for pain scores differed significantly between studies included in the meta-analysis for both NRS 0–10 and 1–10. The findings of this study indicated that comparison of pain scoring used in different studies is difficult due to use of different pain scales and varying nomenclature, and that many variables (such as age and gender) influence pain scoring.

## 1. Introduction

Painful conditions are frequently seen in cattle, caused either by disease or by veterinary or husbandry procedures [1]. Cattle are stoic prey animals; as a result, it is considered that these animals present a higher threshold for pain compared to other species because they show a strong pain-masking behavior [2,3,4]. Therefore, recognition and behavioral changes and categorization of the degree of pain experienced by cattle is the responsibility of both the producer and the veterinarian to preserve a good welfare status [3,5]. However, analgesic treatment to mitigate the level of acute pain is mainly up to the veterinarian [1], but despite this, it is important to note that this treatment is largely dependent on the full knowledge of the normal behavior of the species. Individual animals may differ in their expression of emotions [6]. However, the assessment and evaluation of pain by use of behavioral parameters, such as ethograms [7,8] or facial grimace scales [5], depends on the observer’s experience and attitude [2], and is therefore subjective.

In the last 15 years, many surveys about pain assessment and management in cattle have been published to evaluate attitudes of veterinarians and farmers towards pain and pain management in cattle and assess their use of non-steroidal anti-inflammatory drugs (NSAIDs) in regard to frequency, active components, and occasions [9,10,11]. Studies of pain assessment for different procedures and/or conditions in cattle were conducted among veterinarians [1,9,12] as well as producers [13,14], using either Numerical Rating (NRS) [9,11] or Visual Analogue (VAS) [15] Scales. The concluding results of these studies showed that there is a wide range of attitudes of veterinarians [9,11,12] as well as practitioners [16] about pain assessment in cattle. However, questionnaires about the assessment of pain during procedures and conditions are described to be a promising method to assess the attitudes of participants towards pain in cattle [17].

A NRS is a scale that can be delivered verbally or graphically, and has two end points (“no pain” and “worst pain”) [18]. In bovine medicine, the NRS normally ranges either from 0 (no pain) to 10 (worst pain imaginable) [9,12] or 1 (no pain) to 10 (worst pain imaginable) [1,15]. The VAS is a horizontal line of 100 mm, describing pain limits from “no pain” (0, left side) to worst pain imaginable (10, right side) [15,19]. This scale can also be used in surveys about pain assessment in cattle [15,19,20] and is described to be more informative than the NRS [17].

Scientific evidence shows that in both human and veterinary medicine, there are inherent factors associated with the assessor that can influence the recognition of acute pain, such as the social status, work status, age, gender, degree of empathy, and educational level of the assessor [1,9,12,21,22], as well as inherent factors associated with the animal such as species, age, breed, gender, and even the presence of previous pathologies [23].

However, individual pain scores given for different conditions and procedures throughout the studies show a high variety, from the lowest to the highest score presented to the respondents being selected by individuals [1,9,12].

Numerous studies of pain assessment in cattle have been published [5,24]. However, to this day, there is no systematic review of pain assessment in cattle conducted by veterinarians or producers using different rating scales.

Therefore, the objectives of this systematic review were to (i) describe and compare pain scores and their ranges awarded by veterinarians and producers using either a Numerical Rating or a Visual Analogue Scale, and (ii) compare these scores with a meta-analysis. The aim of this review is to contribute to the current knowledge about pain assessment in cattle, and the possible differences between veterinarians and producers.

## 2. Materials and Methods

### 2.1. Search Strategy and Selection Criteria

The systematic review and meta-analysis were performed following the Preferred Reporting Items for Systematic Reviews and Meta-Analysis protocols (PRISMA-P) study protocols [25] as described by Oehm et al. [26] and Tschoner and Feist [27] (Figure 1, Appendix A). The literature search was conducted using the scientific literature databases PubMed (including MEDLINE) and Agricola on 11 September 2023. A range of years for the analysis of articles was not defined. The search was conducted for all available years. The search terms were separated to include the four components of this review:To identify studies with a study population of veterinarians, farmers, or other people working with cattle: (veterinar* OR farm* OR produc* OR livestock* OR clinic* OR practition* OR caretak*) ANDTo identify studies performed on cattle: (cattle OR cow OR calves OR calf OR dairy OR beef OR bovine) ANDTo identify studies where a questionnaire was used: (survey OR question* OR attitud* OR opinion*) ANDTo identify studies with surveys conducted on pain assessment or management: (pain* OR analges*).

**Figure 1 animals-14-00351-f001:**
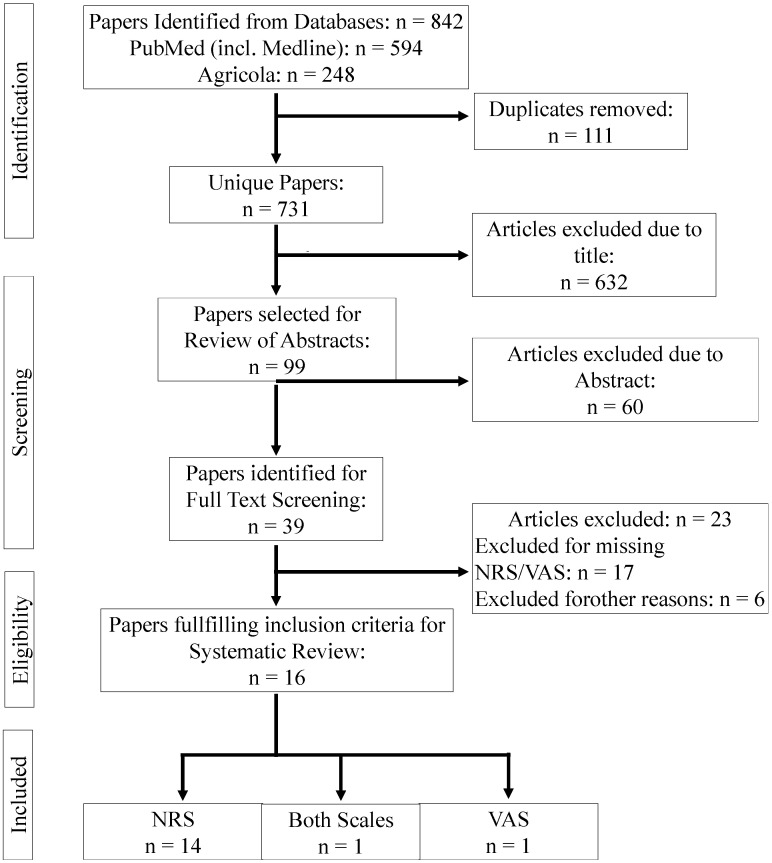
PRISMA (Preferred Reporting Items for Systematic Reviews and Meta-Analysis) flow chart of the literature search and the selection of the studies included in the present systematic review about the comparison of pain assessment using either a Numerical Rating (NRS) or Visual Analogue (VAS) Scale.

Alternative wording was included by using the operator “OR”, and all components were combined by the separator “AND”. Using an asterisk, the databases were screened for words beginning with these letters.

### 2.2. Selection of Studies

Initially, studies of all languages and designs describing pain assessment and/or management by veterinarians, farmers, and producers in cattle were included in the study selection. Subsequently, studies that were not written in German or English, or studies that were not accessible in any way, were not included in this review. De-duplication was conducted manually by the first author (TT) using EndNote (Version X9.3.3). The titles were then screened by TT. Reviews and proceedings, as well as titles including other species than cattle, were excluded at this point. The abstracts of the remaining publications were then evaluated by three authors (TT, KM, MF) to assess whether the eligibility criteria where met. The screening criteria for abstracts were the following:The title and abstract were written in either English or German.The study was conducted using a questionnaire or survey.Veterinarians, farmers, producers, or other people handling cattle were involved.The animal population was cattle.Pain assessment was conducted.

All three reviewers were blinded to the decision of the other reviewers until decisions (include, exclude, maybe) had been made. If a study seemed eligible for two of the three reviewers, the full text was retrieved. All full texts were then screened by TT and were included in the present systematic review if the following questions, as described previously [27,28,29], could be answered with “yes”:(1)Can the full text be obtained?(2)Is the full text written in English or German?(3)Is the study population either veterinarians, producers, or farmers?(4)Is the study design a survey or a questionnaire?(5)Is the questionnaire or survey about the assessment of painful conditions/procedures?(6)Is either a Numerical Rating or Visual Analogue Scale used for pain assessment?(7)Is the questionnaire about cattle?(8)Is the article peer-reviewed?

To objectively compare pain assessment, only full texts including pain scoring using either a NRS or VAS were included. If the screening author was uncertain whether a study should be included, two other authors (KM and MF) were consulted.

### 2.3. Extraction of Data

Data extraction was performed by the first author (TT). Data were extracted regarding primary author, year of publication, country, group, and number of participants, return rate and responses included, demographic data of participants, pain scale used, painful condition and procedures assessed in either adult cows or calves, assessment of necessity and/or use of analgesics, and funding information.

### 2.4. Meta-Analysis and Statistical Analysis

For the meta-analysis, studies with more than three pain scores per condition or procedure were included. Median and mean values as well as SD were collected. If these were not presented in the articles, the first author of the respective article was contacted, with three authors responding and providing the original data. Pain scores were compared with the Kruskal–Wallis test. Studies were divided into articles using NRS 0–10 and NRS 1–10. For the 0 hypothesis, scores of 0 were defined to be no pain. The means of pain scores for professions were compared by the meta-analysis.

## 3. Results

### 3.1. Demographic Findings

A PRISMA flow chart presenting an overview of the literature search and study selection is given in Figure 1. A pool of 842 articles was identified by the search terms in the databases; of these, 111 were duplicates. A total of 731 titles were screened, with 632 excluded at this point due to the title. Abstract screening was conducted for 99 references, with a total of 39 references retrieved for full-text screening, including 3 references for which no abstract was available. Studies for which the abstract was not accessible were included in the full-text retrieval and screening. A total of five references were excluded as they were commentaries (*n* = 2) or books or book chapters with no abstract (*n* = 3). A total of 16 references met all eight inclusion criteria and were therefore included in the systematic review. All abstracts and full texts screened for this systematic review were written in English. The publication year ranged from 2006 to 2022. A total of three studies were conducted in Finland, two studies each in the United Kingdom, New Zealand, Norway, and Germany, and one study in each of Canada, Denmark, Switzerland, Brazil, and China. Demographic information about the articles is presented in Table 1.

### 3.2. Material and Methods

The study design of the articles was heterogenous. The study population was veterinarians in seven studies, farmers in four studies, and either veterinary students, veterinarians and veterinary students, or veterinarians and farmers in one study, respectively. In one study each, the study population was either veterinarians, frontline staff, and managers, or veterinarians, farmers, and claw trimmers. Pain scores were given under the assumption that no analgesia was provided for 11 studies. A total of four studies did not define if pain scoring was conducted under the assumption of pain relief, and one study asked for pain scoring depending on the presence or absence of analgesia (Table 1). Surveys were conducted exclusively in paper form for eight studies and online for five studies. A total of four studies used both paper and online surveys. Pain assessment was presented for procedures and conditions in either adult cattle or in calves in two studies each, and for both in eleven studies. For one of the surveys, the age category of animals was not stated. A total of 14 studies were exclusively about pain management in cattle, whereas one study each also included horses, or horses and pigs, in the survey. NRS ranging from 0 to 10 was used in eight studies, NRS ranging from 1 to 10 in six studies, VAS ranging from 0 to 10 in one study, and both NRS (1–10) and VAS (0–10) in one study (Table 1). In one study, use of NRS (0–10) was described in the Material and Methods section, but VAS (0–10) was used in the Results section. In nine studies, sections about availability and use of analgesics, and/or questions regarding pain management, were included in the survey. Detailed information about Material and Methods is presented in Appendix A.

### 3.3. Funding Information

Funding information was provided for 87.5% (*n* = 14) of studies and is presented in Appendix A.

### 3.4. Pain Scores for Adult Cattle

Pain Scores awarded for different procedures and conditions in adult cattle are presented in Table 2 (procedures) and Table 3 (conditions). The procedures that were rated to be most painful were caesarean section (median 9 for [1,9,11,12,14]), claw amputation (median 10 for [1,9,12,30] and 9 for [11,14]), extirpation of the eye bulb (median 9 for [11] and 10 for [14]), laparotomy (median 9 for [14]), left displacement of the abomasum (LDA) surgery (median 9 for [9,12]), omentopexy (median 9 for [14]), and treatment of interdigital hyperplasia (median 9 for [30]). The conditions and diseases that were rated most painful were acute toxic (*E. coli*) mastitis (median 9 for [13]), dystocia (mean and SD 9.0 ± 1.83 for [16]), and fracture of tuber coxae (median 9 for [15]). Numerical data were not presented for *n* = 3 (18.8%) of the included articles [31,32,33]. Lorena et al. [31] stated that fracture repair was considered the most painful procedure. Pain scores including ranges are given in Appendix A (procedures) and Appendix A (conditions).

### 3.5. Pain Scores for Calves

Pain scores awarded for different procedures and conditions in calves are presented in Table 4 (procedures) and Table 5 (conditions). The procedures considered to be most painful were castration (median of 9 both for Burdizzo and surgical [11,14]), disbudding (median 9 for [20] and mean and SD 9.0 ± 1.2 for [22]), laparotomy (median 9 for [11,14]), repair of distal limb fracture (median 10 for [12]), and umbilical hernia surgery (median 9 for [11,14]). The conditions that were rated to be most painful were ileus (median 9 for [11,14]) and distal limb fracture (mean and SD 9.0 ± 1.2 for [22]). Numerical data were not presented for n = 3 (18.8%) articles [31,32,33]. Pain scores including ranges are given in Appendix A (procedures) and Appendix A (conditions).

### 3.6. Differences between Veterinarians, Farmers, and Others

A comparison of pain scoring between veterinarians and farmers was presented in 25% (*n* = 4) of papers. According to Thomsen et al. [13], farmers considered diseases to be more painful than veterinarians. These differences were significant for left displaced abomasum (*p* < 0.0001), digital dermatitis (*p* = 0.01), mastitis (*p* < 0.001), and interdigital necrobacillosis (*p* < 0.0001). Becker et al. found significant differences in pain perception for therapeutic trim of a sole ulcer and treatment for white-line disease between veterinarians, farmers, and claw trimmers [30]. Contrary to that, pain scores given by Bavarian veterinarians and farmers did not differ significantly [14]. Profession of the participants also had no effect on pain scores in a Chinese study [16].

### 3.7. Influence of Gender on Pain Scoring

Comparison of pain scoring between male and female participants was conducted for 11 studies. A total of three studies [16,19,22] found no differences in pain scoring between genders. In a study from 2006, British female veterinarians awarded significantly higher pain scores for treatment of a sole ulcer (*p* < 0.001), dystocia (*p* < 0.001), fracture of tuber coxae (*p* < 0.001), left displaced abomasum (*p* < 0.001), acute metritis (*p* < 0.001), swollen hock (*p* < 0.001), acute toxic *E. coli* mastitis (*p* < 0.001), mastitis (clots in milk only, *p* < 0.001), and neck calluses (*p* < 0.001) in adult cattle, and surgical castration (*p* < 0.001), disbudding (*p* < 0.001), following dystocia (*p* < 0.01), umbilical abscess (*p* < 0.001), joint ill (*p* < 0.001), and pneumonia (*p* < 0.001) in calves. Male veterinarians ranked claw amputation (*p* < 0.01) and dehorning (*p* < 0.01) as significantly more painful than female veterinarians [9]. Ten years later, Remnant et al. [1] found that female gender of respondents was associated with an increase of 0.36 in pain scoring. According to Laven et al. [12], the Mann–Whitney mean rank score for pain scores was higher for female than male veterinarians from New Zealand, with significant differences for treatment of a white-line abscess, acute metritis, swollen hock, acute toxic mastitis, and white-line disease with sub-sole abscess in adult cattle, and castration (Burdizzo), umbilical abscess, joint ill, and pneumonia in calves (*p* < 0.01, respectively). Pain scores also differed significantly (*p* < 0.05) between Brazil veterinarians, with female veterinarians awarding higher pain scores than male ones for all procedures, expect laparotomy and fractures [31]. Female veterinary students gave median scores that were 0.9 points higher than those of their male colleagues [15]. Bavarian female veterinarians awarded significantly higher pain scores for fetotomy and removal of retained fetal membranes (*p* < 0.01, respectively) [11], whereas Bavarian female farmers scored treatment of interdigital hyperplasia, dehorning, laparoscopic fixation of left displaced abomasum, laparotomy, caesarean section, artificial insemination (*p* = 0.01, respectively), and fetotomy (*p* < 0.01) in adult cattle, and laparotomy in calves (*p* = 0.01) significantly higher [14]. Female veterinarians from New Zealand scored supernumerary teat removal (*p* = 0.009) and disbudding (*p* = 0.003) significantly higher [33] than their male counterparts. However, Kielland et al. [19] found no differences in pain scoring between male and female Norwegian farmers.

### 3.8. Influence of Age on Pain Scoring

According to Huxley and Whay [9], pain scoring differed significantly between British veterinarians who had qualified in different decades, with higher pain scores awarded for dystocia, fracture of tuber coxae, left displaced abomasum, acute metritis, swollen hock, toxic *E. coli* mastitis, mastitis, and neck calluses in adult cattle, and umbilical hernia surgery, umbilical abscess, joint ill, and pneumonia in calves, by veterinarians who had qualified more recently; veterinarians who had been qualified longer awarded significantly higher pain scores for claw amputation and dehorning in adult cattle, and disbudding in calves. Ten years later, British veterinarians graduating before 1990 awarded pain scores that were 0.48 points lower compared with veterinarians graduating since 2010. There was no significant difference in graduation between these years for either group [1].

In a survey from New Zealand, highest median pain scores for 14 out of 24 procedures and conditions were scored by respondents who graduated from 2000 onwards. However, decade of graduation was only associated with a significant (*p* < 0.01) difference in pain scoring for 4 of these 24 conditions and procedures [12]. However, according to Kielland et al. [19], there was no influence of age on median pain scoring in Norwegian farmers.

### 3.9. Influence of Education and Experience on Pain Scoring

Huxley and Whay [9] found that British veterinarians with postgraduate training or qualification assigned significantly higher (*p* ≤ 0.01) pain scores for LDA surgery, and higher pain scores for claw amputation, caesarean section, digital dermatitis, and fracture of a distal limb (*p* ≤ 0.05). A larger amount of time spent working with cattle than other species resulted in significantly lower pain scores for cattle diagnosed with LDA (*p* ≤ 0.01), and significantly higher pain scores for DD (*p* ≤ 0.001). Ten years later, significant differences in pain scoring were observed between year and school of graduation, background of participants prior to university (*p* < 0.01), and holding clinical postgraduate qualifications (*p* < 0.05) [1]. In Bavaria, veterinarians with a graduation date between 1960 and 1970 assigned lower pain scores for 11 out of 33 diseases, and 2 out of 20 procedures, but higher pain scores for claw amputation and dehorning in adult cattle, and surgical castration, tenotomy of contracted tendons, and dehorning in calves [11]. Van Dyke et al. [33] found a significant effect of years since graduation on pain scoring, with male veterinarians awarding lower pain scores with increasing years since graduation, whereas pain scoring was consistent within the group of female veterinarians over the years. Other studies found no influence of level of education [19,22] or experience as a veterinarian [22] on pain scoring.

### 3.10. Results of the Meta-Analysis

A total of eleven articles were included in the meta-analysis: six for NRS 0–10 and five for NRS 1–10. For NRS 0–10, pain scores of 16 procedures (*n* = 10 for cattle and n = 6 for calves) and 7 conditions (*n* = 3 for cattle and *n* = 4 for calves) were compared. For NRS 1–10, pain scores of five procedures and ten conditions in cattle were compared; a total of three conditions (treatment of interdigital hyperplasia, treatment of sole ulcer, treatment of white-line abscess) were excluded, as no SD was available for the pain scores.

Kruskal–Wallis tests showed pain scores were not significantly different between professions (veterinarians and veterinary students, farmers, and practitioners (including frontline staff), neither for NRS 0–10 (*p* = 0.42) nor for NRS 1–10 (*p* = 0.33). The meta-analysis of professions shows a very high heterogeneity within professions, but not significant differences in means between professions. For NRS 0–10, heterogeneity of mean values of pain scores was significant for all procedures and conditions in both calves and cattle (*p* < 0.01, *p* = 0.02 for digital dermatitis, respectively). Forest plots of the meta-analysis for NRS 0–10 are given in Figure 2 (procedures for cattle), Figure 3 (conditions for cattle), Figure 4 (procedures for calves), and Figure 5 (conditions in calves).

For NRS 1–10, heterogeneity of mean values of pain scores was significant (*p* < 0.01) for acute metritis, acute toxic (*E. coli*) mastitis, neck calluses, and swollen hock (Figure 6). For five procedures (claw amputation, dehorning, treatment of sole ulcer, interdigital hyperplasia, and white-line abscess) and one condition (dystocia), the *p*-values for heterogeneity could not be calculated due to missing SD (claw amputation, dehorning, treatment of sole ulcer, interdigital hyperplasia, and white-line abscess).

**Table 1 animals-14-00351-t001:** Demographic information of 16 articles included in the systematic review. All articles were written in English and published in peer-reviewed journals. Articles were included if pain assessment was conducted by use of a survey, and either a Numerical Rating (NRS) or a Visual Analogue (VAS) Scale.

Year	Author	Country	Ref. ^1^	Participants	Return Rate (%)	Responses Included	Gender	Rating Scale	Analgesia ^2^
2006	Huxley and Whay	United Kingdom	[9]	Veterinarians	26.8	615/641	72.6% male, 27.4% female	NRS (0–10)	No
2007	Hewson et al.	Canada	[10]	Veterinarians	50.1	585/586	65% male, 35% female	NRS (1–10)	No
2009	Kielland et al.	Norway	[15]	Veterinary Students	57 ^3^	171/171 ^4^	19.9% male, 80.1% female	VAS (0–10) NRS (1–10)	Not Stated
2009	Laven et al.	New Zealand	[12]	Veterinarians	37	166/166	62.7% male, 37.3% female	NRS (0–10)	No
2010	Kielland et al.	Norway	[19]	Dairy Farmers	70	149/154	87% male, 13% female	VAS (0–10)	Not Stated
2012	Thomsen et al.	Denmark	[13]	Veterinarians Dairy Farmers	28 47	137/493 189/401	Not Stated	NRS (1–10)	No
2013	Becker et al.	Switzerland	[30]	Veterinarians Claw Trimmers Dairy Farmers	Not stated	137 32 77	77.4% male, 22.6% female 100% male 89.6% male, 10.4% female	NRS (1–10)	No
2013	Lorena et al.	Brazil	[31]	Veterinarians	Not Stated	713/800	60% male, 40% female	NRS (1–10)	No
2013	Wikman et al.	Finland	[20]	Dairy Farmers	45	439/451	Not stated ^5^	NRS (0–10) ^6^	Not Stated
2014	Norring et al.	Finland	[22]	Veterinarians Veterinary Students	about 40% about 40% ^7^	189 in total	9% male, 91% female	NRS (0–10)	Not Stated
2015	Hokkanen et al.	Finland	[32]	Veterinary Students	45	438/451	Not Stated	NRS (0–10)	No
2017	Remnant et al.	United Kingdom	[1]	Veterinarians	16 ^8^	242/247	56% male, 44% female	NRS (1–10)	No
2020	Tschoner et al.	Germany	[11]	Veterinarians	26.2	274/287	82.1% male, 17.5% female	NRS (0–10)	No
2021	Tschoner et al.	Germany	[14]	Dairy Farmers	15.4	492/577	79.5% male, 18.7% female	NRS (0–10)	No
2021	Van Dyke et al.	New Zealand ^9^	[33]	Veterinarians	17.6	104/106	48% male, 52% female	NRS (1–10)	No and Yes ^10^
2022	Shi et al.	China	[16]	Veterinarians Frontline Staff	24.1	465/666	90.1% male, 9.9% female	NRS (0–10)	No

^1^ Reference; ^2^ pain assessment was either conducted under the assumption that no analgesia or anesthesia was administered; ^3^ 54.7 for VAS; 59.3 for NRS; ^4^ 82 responses for VAS; 89 responses for NRS; ^5^ 255 men and 175 women, 9 not clarified; ^6^ NRS explained in Material and Methods, but VAS (0–10) used in Result section; ^7^ approximately 42% of students from preclinical and clinical stage, respectively; ^8^ no return rate for online survey, as extent of distribution unknown; ^9^ survey conducted among veterinarians in New Zealand, affiliation of authors in United Kingdom; ^10^ presence or absence or local anesthesia and/or postoperative analgesia.

**Table 2 animals-14-00351-t002:** Pain scoring for different procedures presented to participants of surveys about pain assessment in adult cattle, under the assumption that no analgesics are used. Use of analgesics was not stated for [15,19,20,22]. Pain scoring was conducted by either veterinarians (V), veterinary students (VS), farmers (F), practitioners (and frontline staff, P), and/or claw trimmers (C) by use of a Numerical Rating (NRS, ranging either from 0 to 10 or 1 to 10) or Visual Analogue Scale (VAS, ranging from 0 to 10), or both. Ref. [20] described using a NRS in the Material and Methods section, but indicated use of a VAS in the Results section, and is therefore included as VAS. Left displacement of the abomasum is abbreviated as LDA. If procedures were not presented in the respective reference, this is indicated as -.

	NRS (0–10)						NRS (1–10)								VAS (0–10)		
	[9]	[12]	[22] ^1^	[11]	[14]	[16] ^1^	[10] ^1^	[15]	[13] ^2^		[30]			[1]	[15]	[19]	[20]
Professional Group	V	V	V/VS	V	F	V/F/P	V	VS	V	F	V	F	C	V	VS	F	P
**Procedures on the Head**																	
Dehorning ^3^	8	8	-	8	8	7.1 ± 2.52	7.4	4	-	-	-	-	-	8	3	5.1	-
Extirpation of eye bulb	-	-	-	9	10	-	-	-	-	-	-	-	-	-	-	-	-
**Abdominal Surgeries**																	
Laparoscopic fixation of LDA	-	-	-	-	8	-	-	-	-	-	-	-	-	-	-	-	-
Laparotomy	-	-	-	8	9	-	-	-	-	-	-	-	-	-	-	-	-
LDA surgery	9	9	-	-	-	-	-	-	-	-	-	-	-	8	-	-	-
Omentopexy	-	-	-	-	9	-	-	-	-	-		-	-	-	-	-	-
**Orthopedics**																	
Claw amputation	10	10	-	9	9	-	-	-	-	-	10	10	10	10	-	-	-
Debriding of a digital dermatitis lesion	6	-	-	-	-	-	-	-	-	-	-	-	-	7	-	-	-
Treatment of interdigital hyperplasia ^4^	-	-	-	8	7	-	-	-	-	-	8	8	9	-	-	-	-
Treatment of a sole ulcer ^4^	6	-	-	7	7	-	-	-	-	-	8	7	7	7	-	-	-
Treatment of white-line abscess ^4^	-	4	-	-	-	-	-	-	-	-	8	7	7	-	-	-	-
**Obstetrics and Gynaecology**																	
Artificial Insemination	-	-	-	1	0	-	-	-	-		-			-	-	-	-
Caesarean section	9	9	-	9	9	8.6 ± 2.12	8	-	-		-			9	-	-	-
Fetotomy	-	-	-	7	7/8	-	-	-	-		-			-	-	-	-
Rectal examination	-	-	-	1/2	1	-	-	-	-		-			-	-	-	-
Removal of retained fetal membranes	-	-	-	3	5	-	-	4	-		-			-	2	2.4	-
**Other**																	
Needle prick ^5^	-		2.5 ± 1.87	-	-	-	-	-	-		-			-	-	-	-

^1^ Mean values including standard deviation where indicated; ^2^ median as well as mean values presented in article, median values were included in the table; ^3^ horns > 8 cm long for Huxley and Whay (2006) [9], Laven et al. (2009) [12]; in cattle over 6 months of age for Hewson et al. (2007) [10]; ^4^ excision for Becker et al. (2013) [30], ^5^ fully grown cattle, into the shoulder muscle for Norring et al. (2014) [22].

**Table 3 animals-14-00351-t003:** Pain scoring for different conditions presented to participants of surveys about pain assessment in adult cattle, under the assumption that no analgesics are used. Use of analgesics was not stated for [15,19,20,22]. Pain scoring was conducted by either veterinarians (V), veterinary students (VS), farmers (F), and/or practitioners (and frontline staff, P) by use of a Numerical Rating (NRS, ranging either from 0 to 10 or 1 to 10) or Visual Analogue Scale (VAS, ranging from 0 to 10), or both. Ref. [20] described using a NRS in the Material and Methods section, but indicated use of a VAS in the Results section and is therefore included as VAS. Retained fetal membrane is abbreviated as RFM. If procedures were not presented in the respective reference, this is indicated as -.

	NRS (0–10)						NRS (1–10)					VAS (0–10)		
	[9]	[12]	[22] ^1^	[11]	[14]	[16] ^1^	[10] ^1^	[15]	[13] ^2^		[1]	[15]	[19]	[20]
Professional Group	V	V	V/VS	V	F	V/F/P	V	VS	V	F	V	VS	F	P
**Conditions of the Head**														
Corneal ulcer	-	-	-	-	-	-	5.5	-	-	-	-			
Fracture of the horn	-	-	-	6	6	-	-	-	-	-	-			
Loss of nose ring	-	-	-	6	6	-	-	-	-	-	-			
Neck calluses	2	-	-	3	3	-	-	4	3	4	3	4		
Uveitis ^3^	6	-	-	5	5	-	-	6	8	4	6	5		
**Conditions of the Abdomen**														
Abomasal displacement	-	-	7.3 ± 1.9	-	-	7.4 ± 2.18	-	-	-		-	-	-	8
Left displaced abomasum	3	6	-	5	5	-	-	6	5	6	-	4	7	-
Oesophageal obstruction	-	-	-	-	-	5.9 ± 2.36	-	-	-		-	-	-	-
Right displaced abomasum	-	-	-	6	6	-	-	-	-		-	-	-	-
Ruminal acidosis	-	-	-	-	-	5.3 ± 2.62	-	-	-		-	-	-	-
Severe tympany in cattle ^4^	-	-	7.9 ± 1.6	-	-	6.1 ± 2.18	-	-	-		-	-	-	9
Traumatic pericarditis	-	-	-	-	-	7.8 ± 2.25	-	-	-		-	-	-	-
**Orthopedic Conditions**														
Decubitus	-	-	-	4/5	4/5	-	-	-	-		-	-	-	-
Digital Dermatitis	6	-	-	7	7	-	-	-	7	7	6	-	-	-
Footrot	-	5	-	-	-	-	-	-	-		-	-	-	-
Fracture of long bone ^5^	-	-	-	8	8	8.4 ± 2.08	-	-	-		-	-	-	-
Fracture of tuber coxae ^6^	7	8	-	-	-	-	-	9	8	8	8	8	-	-
Hock with hair loss	3	-	-	-	-	-	-	-	-		3	-	-	-
Hoof disease	-	-	-	-	-	6.9 ± 2.18	-	-	-		-	-	-	-
Injuries on hock ^7^	-	-	-	-	-	-	-	4	-		-	4	2.9	-
Interdigital necrobacillosis	-	-	-	-	-	-	-		8	8	-	-	-	-
Laminitis	-	-	-	8	8	-	-	7	-		-	7	5.7	-
Rupture of muscle	-	-	-	8	8	-	-	-	-		-	-	-	-
Septic Arthritis/Polyarthritis	-	-	-	8	8	-	-	-	-		-	-	-	-
Sole ulcer	-	-	-	8	8	-	-	7	-		-	6	7.1	-
Swollen hock	5	6	-	-	-	-	-	5	5	5	5	5	-	-
White-line disease ^8^	7	-	-	-	-	-	-	-	-		7	-	-	-
**Mastitis and Udder Health**														
Acute mastitis ^9^	-	-	7.3 ± 1.4	-	-	-	-				-	-	7.6	8
Acute toxic (E. Coli) mastitis ^10^	7	8	-	7	7	7 ± 2.2	-	7	9	9	7	7	-	-
Intertrigo	-	-	-	4	4	-	-	-	-		-	-	-	-
Mastitis (clots in milk only) ^11^	3	3	-	1	1	3.4 ± 2.65	-	4	2	3	4	5	-	-
Moderate mastitis	-	-	-	-	-	5.1 ± 2.24	-	-	-		-	-	-	-
Open teat injury	-	-	-	6	6	-	-	-	-		-	-	-	-
Teat injury ^12^	-	-	7.4 ± 1.7	-	-	-	-	-	-		-	-	-	8
**Obstetrics and Gynaecology**														
Acute metritis ^13^	4	-	-	5	5	-	-	7	6	6	5	4	-	-
After removal of RFM	-	-	-	-	-	-	-	4	-		-	4	2.4	-
Calving	-	-	-	-	-	8.5 ± 1.99	-		-		-	-	-	-
Dystocia ^14^	7	7	7.3 ± 1.7	8	8	9.0 ± 1.83	5.3	8	-		7	8	-	-
Endometritis	-	-	-	-	-	5.9 ± 2.62	-	-	-		-	-	-	-
Postpartum paralysis	-	-	-	-	-	5.9 ± 3.02	-	-	-		-	-	-	-
Tissue injuries following birth	-	-	-	5	5	-	-	-	-		-	-	-	-
Uterine torsion	-	-	-	6	6	-	-	-	-		-	-	-	-
Uterine prolapse ^15^	-	-	6.9 ± 2.0	5	5	7.9 ± 2.3	-	-	-		-	-	-	8
Vaginal prolapse	-	-				6.3 ± 2.66	-	-	-		-	-	-	-
**Metabolic and Nutritional Diseases**														
Hypocalcemia ^16^	-	-	-	1	1	-	-	5	-		-	5	3.3	-
Ketosis	-	-	-	1	1	-	-	4	-		-	4	-	-
Nutritional deficiency disease	-	-	-	-	-	3.6 ± 2.8	-	-	-		-	-	-	-
**Other**														
Infectious disease	-	-	-	-	-	4.5 ± 2.91	-	-	-		-	-	-	-
Parasitic disease	-	-	-	-	-	4.1 ± 2.57	-	-	-		-	-	-	-

^1^ Mean values including standard deviation where indicated; ^2^ median as well as mean values presented in reference, median values were included in the table; ^3^ eye infection for Kielland et al. (2009; 2010) [15,19]; ^4^ ruminal bloat for Shi et al. (2022) [16]; ^5^ fracture for Shi et al. (2022) [16]; ^6^ one-sided for Kielland et al. (2010) [19]; ^7^ skin lesions on hock for Kielland et al. (2010) [19]; ^8^ with subsole abscess for Huxley and Whay (2006) [9]; ^9^ fever 41 °C, lumps in milk, hard udder for Norring et al. (2014) [22]; ^10^ *Escherichia coli* mastitis for Huxley and Whay (2006) [9], serious mastitis for Kielland et al. (2009; 2010) [15,19]; severe mastitis for Shi et al. (2022) [16]; ^11^ mastitis for Thomsen et al. (2012) [13], mild mastitis for Shi et al. (2022) [16], chronic mastitis for Tschoner (2020; 2021) [11,14]; ^12^ teat tramping in cows for Norring et al. (2014) [22], teat broken at the roof for Wikman et al. (2013) [20]; ^13^ puerperal metritis for Tschoner et al. (2020, 2021) [11,14], metritis for Remnant et al. 2017 [1]; ^14^ fetal-maternal disproportion requiring traction alone for Huxley and Whay (2006) [9], Laven et al. (2009) [12], Tschoner et al. (2020, 2021) [11,14]; strong pull assistance for Norring et al. (2014) [22]; ^15^ uterine eversion for Tschoner et al. (2020, 2021) [11,14]; ^16^ milk fever for Kielland et al. (2009; 2010) [15,19].

**Table 4 animals-14-00351-t004:** Pain scoring for different procedures presented to participants of surveys about pain assessment in calves, under the assumption that no analgesics are used. Use of analgesics was not stated for [15,19,20,22]. Pain scoring was conducted either by veterinarians (V), veterinary students (VS), farmers (F), and/or practitioners (and frontline staff, P) by use of a Numerical Rating (NRS, ranging either from 0 to 10 or 1 to 10) or Visual Analogue Scale (VAS, ranging from 0 to 10), or both. Ref. [20] described using a NRS in the Material and Methods section, but indicated use of a VAS in the Results section and is therefore included as VAS. Ranges are included in brackets if indicated in the references.

	NRS 0–10						NRS (1–10)			VAS (0–10)		
	[9]	[12]	[22] ^1^	[11]	[14]	[16] ^1^	[10] ^1^	[15]	[1]	[15]	[19]	[20]
Professional Group	V	V	V/VS	V	F	V/F/P	V	VS	V	VS	F	P
**Castration**												
Castration ^2^ up to 6 months	-	-	-	-	-	-	4.9	-	-	-	-	-
Castration ^2^ over 6 months	-	-	-	-	-	-	5.9	-	-	-	-	-
Castration (Burdizzo)	7	6	-	9	9	-	-	-	6	-	-	-
Castration (Rubber Ring)	6	5	-	-	-	-	-	-	6	-	-	-
Castration (Surgical)	6	8	-	9	9	7.8 ± 2.32	-	-	7	-	-	-
**Dehorning/Disbudding**												
Dehorning ^3^	-	-	-	8	8	-	6.8	-	-	-	-	-
Dehorning over 6 months	-	-	-	-	-	-	7.4	-	-	-	-	-
Disbudding	7	8	9 ± 1.2	-	-	7.6 ± 2.32	-	-	7	-	-	9
Disbudding (caustic paste)	-	-	-	-	-	5.6 ± 2.52	-	-	-	-	-	-
Disbudding with analgesics ^4^	-	-	2.4 ± 1.8	-	-	-	-	-	-	-	-	-
**Abdominal Surgery**												
Laparotomy	-	-	-	9	9	-	-	-	-	-	-	-
Umbilical hernia surgery ^5^	8	8	-	9	9	6.8 ± 2.32	7.3	-	8	-	-	-
**Orthopedic Procedures**												
Repair of distal limb fracture	-	10	-	-	-	-	-	-	-	-	-	-
Tenotomy of contracted tendons	-	-	-	8	8	-	-	-	-	-	-	-
**Other**												
Ear tagging	-	-	-	4	4	-	-	-	-	-	-	-

^1^ Mean values including standard deviation where indicated; ^2^ Hewson et al. (2007) [10] did not distinguish between methods of castration; ^3^ in calves up to 6 months for Hewson et al. (2007) [10]; ^4^ pain during burning for Wikman et al. (2013) [20] and Norring et al. (2014) [22]; ^5^ in calves up to 3 months for Hewson et al. (2007) [10].

**Table 5 animals-14-00351-t005:** Pain scoring for different conditions presented to participants of surveys about pain assessment in calves, under the assumption that no analgesics are used. Use of analgesics was not stated for [15,19,20,22]. Pain scoring was conducted either by veterinarians (V), veterinary students (VS), farmers (F), and/or practitioners (and frontline staff, P) by use of a Numerical Rating (NRS, ranging either from 0 to 10 or 1 to 10) or Visual Analogue Scale (VAS, ranging from 0 to 10), or both. Ref. [20] described using a NRS in the Material and Methods section, but indicated use of a VAS in the Results section and is therefore included as VAS. Ranges are included in brackets if indicated in the references.

	NRS 0–10						NRS (1–10)			VAS (0–10)		
Reference	[9]	[12]	[22] ^1^	[11]	[14]	[16] ^1^	[10] ^1^	[15]	[1]	[15]	[19]	[20]
Professional Group	V	V	V/VS	V	F	V/F/P	V	VS	V	VS	F	P
**Abdominal Conditions**												
Enteritis/Diarrhea ^2^	-	-	-	5	5	4.6 ± 2.5	-	5	6	6	4.8	-
Ileus	-	-	-	9	9	-	-	-	-	-	-	-
Ruminal acidosis	-	-	-	4	4	-	-	-	-	-	-	-
**Umbilical Conditions**												
Navel infection	-	-	6.8 ± 1.6	7	7	-	-	5	-	5	5.2	8
Umbilical abscess	5	5	-	-	-	-	-	-	6	-	-	-
Umbilical hernia ^3^	-	-	-	-	-	-	-	-	-	-	-	6
**Orthopedic Conditions**												
Contracted tendons	-	-	-	4	4	-	-	-	-	-	-	-
Joint ill ^4^	7	8	-	8	8	-	-	7	7	6	6.7	-
Distal limb fracture ^5^	8	-	9 ± 1.2	8	8	-	-	8	9	8	7.6	-
**Other**												
(Broncho)Pneumonia	6	8	-	6	6	-	-	6	7	6	6.7	-
Following dystocia ^6^	4	3	5.9 ± 1.9	5	5	-	-	4	5	4	3.3	-
Meningitis	-	-	-	8	8	-	-	-	-	-	-	-
Needle prick neck	-	-	2.4 ± 1.6	-	-	-	-	-	-	-	-	-

^1^ Mean values including standard deviation where indicated; ^2^ intestinal infection for Kielland et al., 2009 [15], no age defined for Shi et al., 2022 [16]; ^3^ the size of an apple for Wikman et al. (2013) [20]; ^4^ septic arthritis/polyarthritis for Tschoner et al. (2020, 2021) [11,14]; ^5^ broken bone with open fracture on calf’s hind leg for Norring et al. (2014) [22], fracture of long bone for Tschoner et al. (2020, 2021) [11,14]; ^6^ fetal-maternal disproportion requiring traction alone for Huxley and Whay (2006) [9], Tschoner et al. (2020, 2021) [11,14], strong pull assistance for Norring et al. (2014) [22].

**Figure 2 animals-14-00351-f002:**
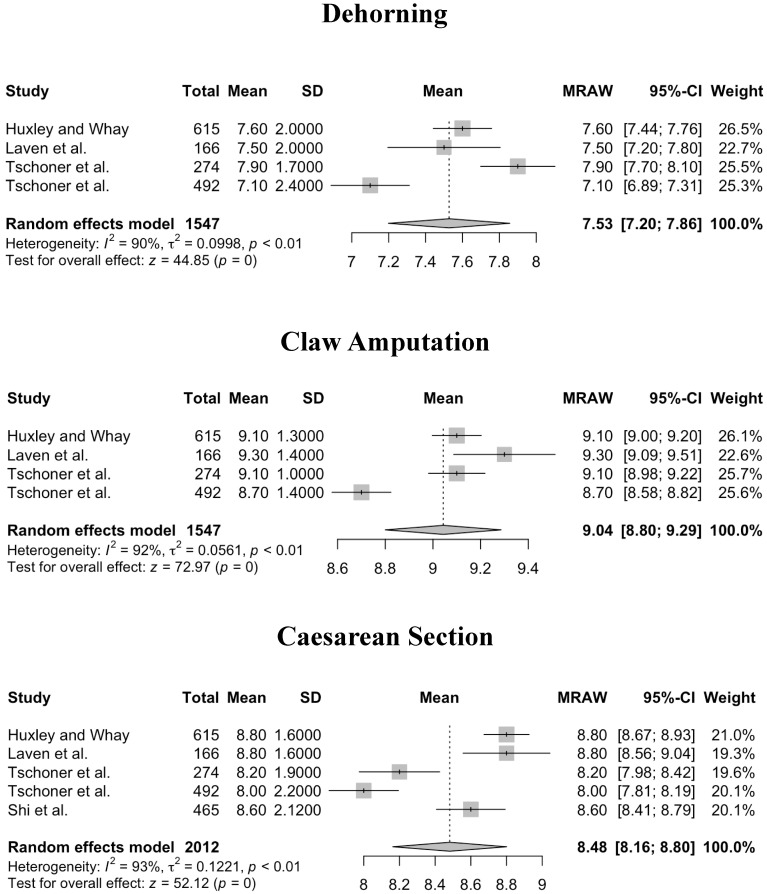
Forest plots for mean and SD of pain scores assigned using a Numerical Rating Scale ranging from 0 to 10 for procedures in cattle. Heterogeneity of mean values of pain scores was significant (*p* > 0.01) for all procedures. Test for overall effect shows a significant pain score measured for cows across studies, where no pain (pain score = 0) is the null hypothesis [9,11,12,14,16].

**Figure 3 animals-14-00351-f003:**
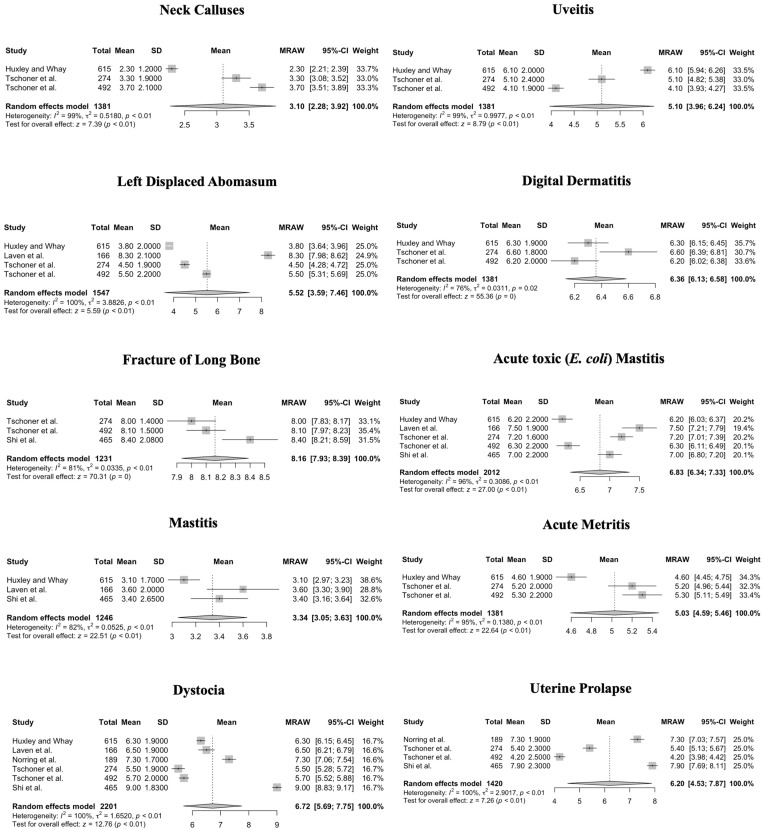
Forest plots for mean and SD of pain scores assigned using a Numerical Rating Scale ranging from 0 to 10 for conditions in cattle. Heterogeneity of mean values of pain scores was significant for all procedures (*p* > 0.01 respectively, *p* = 0.02 for digital dermatitis). Test for overall effect shows a significant pain score measured for cows across studies, where no pain (pain score = 0) is the null hypothesis [9,11,12,14,16,22].

**Figure 4 animals-14-00351-f004:**
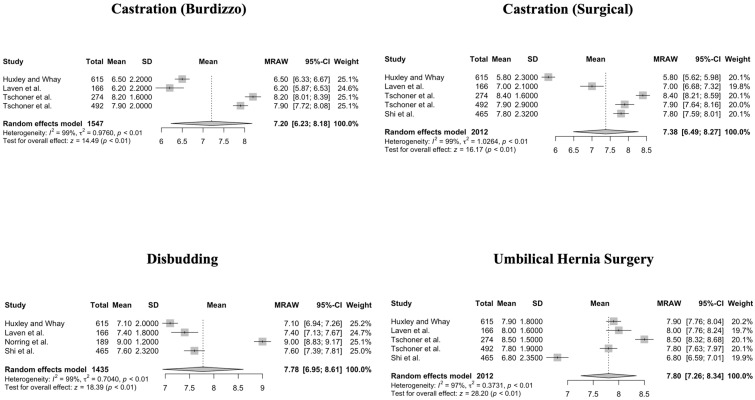
Forest plots for mean and SD of pain scores assigned using a Numerical Rating Scale ranging from 0 to 10 for procedures in calves. Heterogeneity of mean values was significant (*p* < 0.01) for all procedures. Test for overall effect shows a significant pain score measured for cows across studies, where no pain (pain score = 0) is the null hypothesis [9,11,12,14,16,22].

**Figure 5 animals-14-00351-f005:**
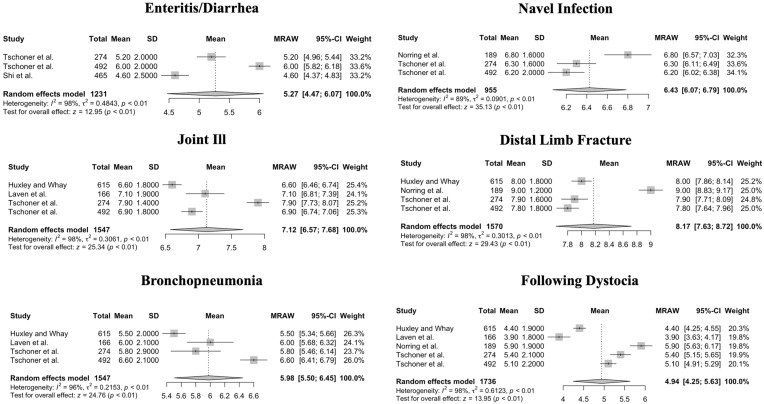
Forest plots for mean and SD of pain scores assigned using a Numerical Rating Scale ranging from 0 to 10 for conditions in calves. Heterogeneity of mean values of pain scores was significant (*p* < 0.01) for all procedures. Test for overall effect shows a significant pain score measured for cows across studies, where no pain (pain score = 0) is the null hypothesis [9,11,12,14,16,22].

**Figure 6 animals-14-00351-f006:**
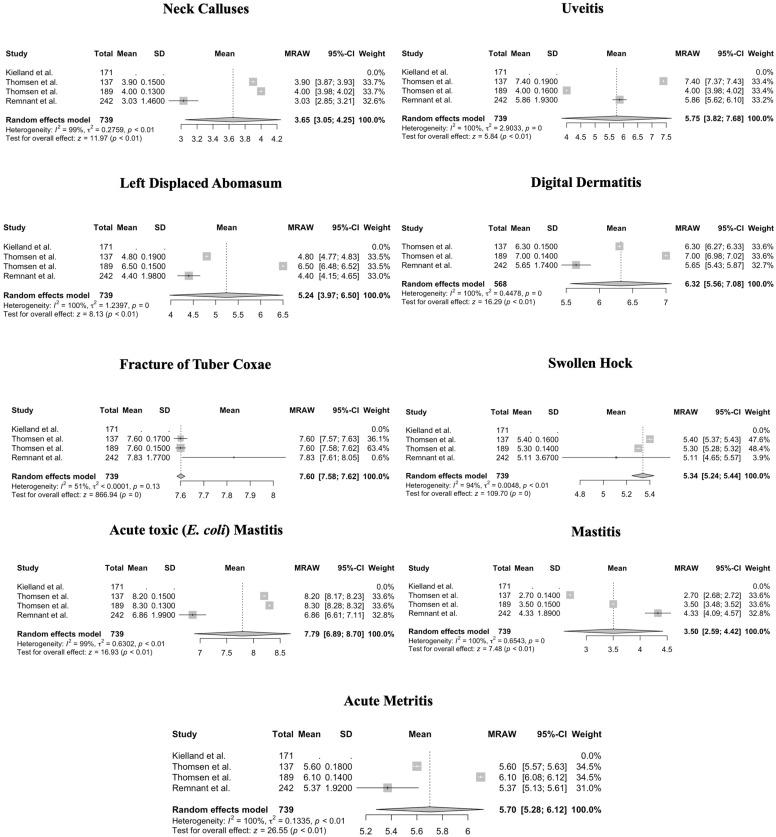
Forest plots for mean and SD of pain scores assigned using a Numerical Rating Scale ranging from 1 to 10 for conditions in cattle. Heterogeneity of mean values of pain scores was significant (*p* < 0.01) for acute metritis, acute toxic (*E. coli)* mastitis, neck calluses, and swollen hock. Test for overall effect shows a significant pain score measured for cows across studies, where no pain (pain score = 0) is the null hypothesis [1,13,15].

## 4. Discussion

### 4.1. Findings of the Systematic Review

The objective of the present systematic review was to describe and compare pain scores assigned by veterinarians, veterinary students, and farmers to different procedures and conditions in cattle. We wanted to assess the existing body of research, indicating areas where knowledge could be increased. Pain assessment is influenced by sex and age, and there are significant differences in pain scores given for some painful procedures and conditions between countries.

Even with a large number (*n* = 842) of articles extracted from the databases, there was a manageable number (*n* = 99) of references left after removal of duplicates and screening of titles, resulting in a small number (*n* = 16) of references included in this systematic review after the full-text screening. This provides evidence that research about pain assessment in cattle using NRS or VAS is still rare. Out of the 39 articles retrieved for full-text screening, 13 were only excluded due to not working with a NRS or VAS but would have otherwise fulfilled all other criteria. This is a limitation of the present study, as pain assessment was conducted in these articles, even if not by using NRS or VAS, but agreement with either “Yes/No” [34] or other predetermined statements [35,36], or pain scales ranging from five (e.g., “not important” to “extremely important” [37], “not painful” to “severe pain”, including “cannot assess” [38]) to six (”no pain” to “worst pain imaginable” [39,40]) categories.

Another major limitation of comparing the studies included in this systematic review is the difference in the pain scales that were used, making direct comparison of pain scores between studies impossible. A total of eight studies worked with a NRS ranging from 0 to 10; in seven studies, NRS ranged from 1 to 10; and two studies used either both NRS and VAS, or only the VAS. Kielland et al. (2009) [15] compared median pain scores assigned by veterinary students using either a NRS (1–10) or a VAS (0–10), finding that students assigned a score that was 0.9 higher via NRS, which correlated with the different ranges of the two scales. In veterinary medicine, the VAS is described to be more informative than the NRS [17]. Nevertheless, VAS was only used in 2 out of 16 studies. According to a systematic review, correlation between VAS and NRS is good in human medicine, with some discrepancies depending on the situation [41]. Most literature in human medicine only compares different pain scales, such as NRS, VAS, and others [41,42], and not different scales of NRS. For comparability between studies, use of the same pain scale would be advisable.

Another problem of comparing the median or mean pain scores of the included articles is the different professions of participants. For the majority of studies, the questionnaire was sent to veterinarians, but other studies compared pain scoring between veterinarians and farmers [13], or veterinarians, farmers, and claw trimmers [30]. However, as median or mean pain scoring was provided for each profession in these articles, comparability between professions and with other articles was given. One other study combined groups of professions, e.g., veterinarians and veterinary students [22], which could have influenced the pain scoring, as veterinary students have less experience with managing animals in pain and with assessing painful conditions and procedures. Another study combined different professions (frontline staff, managers, veterinarians working with cattle) as dairy practitioners [16]. Research shows that pain perception differs between farmers and veterinarians [13], as well as between veterinarians, farmers, and claw trimmers [30]. Even if profession of Chinese dairy practitioners had no influence on pain scoring [16], and pain scores did not differ significantly between Bavarian veterinarians and farmers, there were differences in the perception of the painfulness of conditions and procedures [11,14]. Therefore, the combination of different professional groups for pain assessment could have an influence on study results, and professional groups should be evaluated individually.

The distribution between participating sexes was different throughout the studies, with a higher proportion of male participants compared with female participants for 10 out of 16 studies. The proportion of female participants was only higher for three studies [15,22,33]. In the two articles distributing the survey among veterinary students, as many as 81% [15] and 91% [22] respondents were female, which could be explained by the fact that nearly 80% of veterinary students are women [43]. According to Irvine et al. [43], about half of practicing veterinarians are female, but women are outnumbered by men in food animal practice, contrary to small animal and equine practice [44], which could explain the uneven distribution of genders throughout the studies. Gender distribution could also be influenced by profession; in surveys distributed to other professions, male participants accounted for 85% [19], 89.6% [30], and 79.5% [14] for Danish, Swiss, and Bavarian farmers, respectively; 100% for Swiss claw trimmers [30]; and 90.1% for Chinese practitioners, which included veterinarians as well as frontline staff [16]. As female participants ranked cattle pain higher [1,9,11,12,31], this imbalance in the gender of the participants could have an influence on the pain scores.

Age and year of graduation seem to have an influence on pain assessment, with higher pain scores awarded by more recent graduates [1,9,12]. The publication range of the articles was from 2006 to 2022. Recognition of pain in cattle has been lagging behind that in companion animals and horses, with pain scoring systems for cattle only published in recent years [5,45], which could explain the higher awareness for pain in cattle by recently graduated veterinarians. Only one study compared pain scores awarded by UK veterinarians to a study published ten years before [1], and found that pain perception in cattle veterinarians increased since the study conducted in 2006 [9], with higher pain scores given to over 40% of the listed procedures and conditions [1]. However, no other studies compared data over a period of time, so no other statement can be made about the development of pain assessment for this systematic review.

Another factor that should be considered is the nationality of participants. In the European Union, there is no species-specific legislation for dairy cattle welfare, except for calves [46,47,48]. Regulatory regimes of countries are not always in accordance with perspectives of veterinarians [33]. Most studies were conducted in Europe, except for five studies originating from Canada [10], New Zealand [12,33], Brazil [31], and China [16]. According to van Dyke et al. [33], demographic effects influence the perceptions of pain management in NZ veterinarians. Therefore, nationality, as well as origin of participants, and different opinions and attitudes towards pain in cattle, could likely have influenced the pain scoring. For example, on a NRS from 0 to 10, umbilical hernia surgery was scored with a median of 8 for UK and NZ veterinarians [9,12], and 9 for Bavarian veterinarians [11], and a mean of 6.8 for Chinese practitioners [16]; on a NRS from 1 to 10, umbilical hernia surgery was scored with a mean of 7.3 for Canadian veterinarians [10].

Nomenclature, as well as procedures and conditions presented to participants of the surveys, was heterogenous. Additionally, definitions of procedures and conditions differed throughout the studies. For example, mastitis was given as either clots in milk only for [9,12,15,19], which was defined as chronic mastitis by [11,14]. Other authors asked for pain assessment for grade 1 mastitis [1], mastitis [13], serious mastitis [15,19], or mild, moderate, and severe mastitis [16]. Another condition for assessment was acute toxic *Escherichia coli* mastitis [9], acute toxic mastitis [1,12], *Escherichia coli* mastitis [13], acute mastitis (*Escherichia coli*) [11,14], or acute mastitis with 41 °C of fever, lumps in milk, and a hard udder [22]. These different definitions of either conditions or procedures make comparison of pain scales throughout studies complicated. Translation of terms from the original language of the survey into English, as well as presenting different terms to farmers and veterinarians, as was done in one study (e.g., laparotomy for veterinarians, and omentopexy of displaced abomasum and laparotomy for farmers) [11], is another factor influencing the uniformity of nomenclature. Especially when including farmers or claw trimmers in the survey, authors might have chosen to use lay terms for conditions and procedures to make sure those can be understood by the participants.

Return rates differed widely between studies, from as high as 70% for Norwegian farmers [19] to 15.4% in Bavarian farmers [14]. It is reasonable to think that people interested in pain management in cattle, as well as empathic people, are more likely to participate in a survey about pain. The level of empathy of a human being towards an animal might be influenced by the species of the animal the observer evaluates. The concern for the welfare of an animal could result from the evolution of the human trait, which can be strongly influenced by culture [49]. Including only participants who responded to the survey is a possible bias. As empathic veterinarians were found to give higher pain scores [22], empathy and attitudes towards pain could have an influence on the findings in the articles included in this systematic review. None of the authors stated if they were working with a reward system for participating in the studies, which is another factor that could have influenced the results.

Only one study included assessment of behavioral and postural parameters used for pain recognition in cattle, showing that veterinarians and farmers differed significantly in the parameters they use for pain assessment for 19 out of 28 parameters presented in the survey [14]. Given the wide variety of pain scores assessed for different procedures and conditions, questions about tools and methods for pain assessment, or evaluation of parameters used to recognize if cattle are in pain, it would have been interesting to determine in studies if there is a lack in education concerning the recognition of pain. Improvement in pain recognition could result in a higher awareness of cattle being in pain, and in higher pain scores given to cattle.

### 4.2. Findings of the Meta-Analysis

Meta-analysis was performed for six articles for NRS 0–10 and five articles for NRS 1–10. This small number of articles included in the meta-analysis is due to the fact that we only included pain scores of procedures and conditions included in at least three articles. For the meta-analysis, mean values and SD of pain scores were used; however, in most studies, median values were presented [9,12], as data were not normally distributed, which was not optimal. If mean values and SD were not given in articles, authors were contacted to collect missing data, but only three authors answered to the mails. Data describing pain scores for NRS 1–10 were difficult to interpret, due to the missing SD for the majority of included studies. Therefore, these results should not be relied on, which is a limitation of the meta-analysis. Even if the study design was somewhat similar throughout the studies, heterogeneity of the articles was large, with significant differences between the mean pain scores for all procedures and conditions included for NRS 0–10. Even though individual studies found differences in pain assessment according to profession [13,30], our meta-analysis found no significant differences between mean pain scores, which is in accordance with other studies [14,16]. As articles were heterogenous and SD was not provided for some articles, and professions differed between articles, the results of the present meta-analysis represent insufficiently strong conclusions. That is why we compared the means of pains scores (without SDs) of professions via Kruskal–Wallis, to complement the meta-analysis. However, the low number of studies for a particular condition and the wide discrepancies in pain scores between professions and articles, even for the same diseases, hint at the lack of knowledge in this area and the need to collect more data and conduct additional research to close the knowledge gap. Moreover, the pain scales themselves should be unified to one scoring system instead of the two ranging either from 0 to 10 or 1 to 10. Such unification of pain scores and clarifying the pain heterogeneity is of a huge practical importance for veterinarians and researchers to be able to compare the pain scores given for different procedures and conditions. Therefore, the recommendation should be for researchers (a third person assessing pain in animals in theory without actually looking at a patient) as well as clinical practice (people assessing pain of a patient in their care) to use the same scales for better comparability.

The benefits of conducting studies about pain assessment are collecting data about evaluation of pain and pain management, and learning about areas where more education is needed. Another benefit could be that the individual participant is working through a pain assessment questionnaire, thinking about painful events in a cow, and realizing the number of painful procedures and conditions in a calf’s or cow’s life, possibly thus exercising empathy towards the animal. As empathic veterinarians score pain in cattle higher [22], increasing empathy towards adult cows and calves could result in improved welfare and pain management for cattle.

### 4.3. Use of Analgesics

The presentation and description of the use of analgesics was so heterogenous in the papers that an assessment and comparison of analgesic use was not possible. There were only ten papers describing analgesic use overall. Huxley and Whay [9] presented the percentage of analgesic classes used for different procedures and conditions, Hewson et al. [10] described the mean percentage of animals (beef and dairy) receiving any kind of analgesia, including the top two active components, and Becker et al. [30] only reported the percentage of respondents stating if a local anesthesia was reasonable for procedures involving the claw. Lorena et al. [31] presented use of different analgesic classes for cattle and horses combined, and Norring et al. [22] asked how many veterinarians and clinical students would use a combination of sedation, local anesthesia, and analgesia for disbudding in calves. Remnant et al. [1] included a figure overlaying respondents using NSAIDs in 50% of cases over the stated pain scores. Tschoner et al. [11] divided the use of different classes of analgesics into categories (regularly, frequently, occasionally, never) and presented the percentage of respondents agreeing to each category, and asked about agreement if local anesthesia and NSAIDs were necessary during and after painful procedures [14]. Van Dyke et al. [33] only included pain management protocols for four procedures, and Shi et al. [16] presented a figure showing the proportion of which analgesics would be used by respondents (multi-response answer). Therefore, the description of the use of analgesics is hard. It is also not possible to compare the use of analgesics between animal species, as this depends on which analgesics are labeled for use in food-producing animals. Questions about analgesics in cattle usually refer to NSAIDs and local anesthetics. Reviews of pain management in cattle have been published [50,51]—thus, pain management will not be discussed here.

### 4.4. Methodology and Limitations

The present systematic review was conducted following the PRISMA guidelines [25], as described previously [26,27]. This was done to reduce the possible risk of bias for the study selection process and analysis. Registration via PROSPERO was not possible, as it can only be used for systematic reviews in human medicine and research [27]. To further reduce the risk of bias, the titles and abstracts were screened independently by three authors, and the full-text screening was undertaken using previously specified guidelines. Eligibility for a meta-analysis was discussed with a statistician, as described previously [26].

### 4.5. Risk of Bias

To reduce the risk of missing any articles, the authors used two search engines. Titles and abstracts were included in the keyword search, which should also result in not missing any relevant articles [27]. All articles included in the systematic search were published in English, and no article had to be excluded due to the language not being English or German. Therefore, a bias because of the language barrier can be excluded. A total of three abstracts could not be accessed; as these articles were then included in full-text retrieval, and all articles selected for full-text screening could be accessed, risk of bias due to limited access can also be excluded. The origin of articles was evenly distributed, with 16 articles originating from 10 countries. Therefore, country of origin should not have influenced the present study. Funding information was provided for thirteen of the sixteen articles, with two articles not stating any funding [15,31], and one receiving no external funding [33]. Pharmaceutical companies were involved in the funding of seven of sixteen articles; however, as the articles focused on pain assessment using pain scales, this is unlikely to have had any influence on the results of the study.

## 5. Conclusions

This systematic review should aid researchers to identify gaps in the current knowledge for conceptualization of objectives and the study design for future research.

Studies of pain assessment using NRS or VAS are rare in bovine medicine, and use of pain scales is heterogenous, making comparison of pain scores difficult. There are many variables possibly influencing pain assessment, such as gender, age, education, or profession. Studies mainly originate from the European Union, and research about pain assessment in other countries should be conducted, or published if they are performed. Researchers should focus on using one pain scale throughout studies for better comparability, since there seem to be no clear benefits of using less common pain scales over the commonly used 0 to 10 NRS scale. Additionally, the nomenclature of terms should be consistent, and pain scoring conducted by different professions should be assessed individually. We recommend researchers assess behavioral and postural parameters used by veterinarians and farmers to assess pain in cattle, to evaluate if pain assessment can be improved by training and education, thus improving dairy cattle welfare. Future studies could compare bovine pain scales, changes in perceptions of pain levels after a period of clinical training of respondents, and coherent assessment of use of analgesics in cattle. Assessment of pain should not be performed under the assumption that no pain medication was given, as this is not feasible anymore, especially for surgical procedures.

## Data Availability

The data lies with the corresponding author.

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
