# Peer review of "Pain Assessment in Cattle by Use of Numerical Rating and Visual Analogue Scales—A Systematic Review and Meta-Analysis"

_animals, 2024, doi:10.3390/ani14020351_

Round 1

Reviewer 1 Report

Comments and Suggestions for Authors

This is an interesting review, that contributes with a thorough analysis about a topic that sure needs more research.

Only general comments:

P3L148: or calves

P4L171: 12.5% correspond to what?

Figure 1: Articles excluded due to abstract n=60…abstract not accessible n=3 and abstract not in English/German n=0? Should be 57 for total 60?

P4L178-181: maybe that phrase is more material and methods than part of a figure title?

P7L296: begin the phrase mentioning the authors instead of the citation number please (same for others).

P8L535: [32]Most…missing a dot and space.

Conclusions: L648: Published studies are mainly originating from Europe…research, in this topic, are performed in other countries, however are not always published.

Reviewer 2 Report

Comments and Suggestions for Authors

General comments

Pain assessment in animals is a crucial event in pain management due to the need to reduce the physiological and behavioral impact of the perception of this sign. However, as shown in the proposal of your manuscript, the influence of different factors associated to the evaluator is limiting and also affects its evaluation in this species as it has been observed in small species. Therefore, I consider that your proposal is innovative and appropriate for this journal. However, a limitation of this manuscript is the lack of approach to current scales that have been proposed that could improve the perspective of acute pain recognition, so I suggest that prior adjustments should be made.

Response:

Particular comments

Lines 11 - 12. I understand the proposal of your simple summary I agree with it, however, due to the suggestion by the authors guide it should be brief, so if the authors allow me I suggest that these sentences would be more appropriate for the beginning of your abstract.

Response:

Line 13. Did the authors consider a range of years for the analysis of articles they included in their study? If so I suggest that they can include in this sentence to describe their methodology briefly.

Response:

Line 29. In complement to my previous comment I suggest that you include the range of years of the articles you included in your study and the databases you considered.

Response:

Line 34. Although this was not an objective of your study, I suggest that you could include which analgesics were used in the articles you reviewed.

Response:

Line 38. Please consider including "acute pain" in your keywords as this may increase your chances of being matched in the different databases.

Response:

Line 44. I agree with your statement, however, if the authors allow me I suggest to include in this sentence "cattle are stoic prey animals, because of this it is considered that these animals present a higher threshold to pain compared to other species because of showing a strong pain masking behavior".

Response:

Lines 44 - 45. I consider these statements to be very important and linked to each other, however, the idea is not very clear. If the authors allow me, I suggest that it could be modified to "Therefore, recognition of behavioral changes and categorization of the degree of pain experienced by cattle is the responsibility of both the producer and the veterinarian to preserve a good welfare status [3][5]. However, analgesic treatment to mitigate the level of acute pain is mainly up to the veterinarian [1], but despite this, it is important to note that this treatment is largely limited on the full knowledge of the normal behavior of the species". I suggest this article could be consulted to complement your references "Mota- Rojas et al. (2022) The Neurobiology of Behavior and Its Applicability for Animal Welfare: A Review doi: https://doi.org/10.3390/ani12070928".

Response:

Line 52. Please describe for the first time the acronyms you have used.

Response:

Line 53. Please add reference.

Response:

Lines 68 - 70. This paragraph I believe presents a crucial idea in your study, however, I suggest that it should be reordered to clearly convey your idea. If the authors agree I recommend that it be modified by "Scientific evidence shows that in both human and veterinary medicine there are inherent factors associated with the assessor that can influence the recognition of acute pain such as social status, work status, age, gender, degree of empathy and educational level of the assessor that influence pain recognition [21,1,8,9,12,22,14]. As well as inherent factors associated with the animal such as species, age, breed, gender and even the presence of previous pathologies [Hugonnard et al. 2004]." please refer to this reference Hugonnard et al. (2004) Attitudes and concerns of French veterinarians towards pain and analgesia in dogs and cats doi: 10.1111/j.1467-2987.2004.00175.x, which might help to complement your references.

Response:

Line 71. Please add references that support the idea about findings in different countries.

Response:

Line 77. As a general recommendation on these statements if you claim that different studies have reported it I suggest that you add these references.

Response:

Lines 88 - 89. I suggest that this sentence be adequate because the presence of the quotes in between does not give continuity to the reading, therefore I suggest that they be relocated to the end of the sentence or split into two different sentences. Likewise when they mention the phrase "As described by [24,25]" be replaced by "As described by Oehm et al. [24] and Tschonet et al. [25]".

Response:

Line 91. In complementation with my comment made on line 29, the authors could mention the range of years that were considered for the collection of the articles within the review. Also at this same level I suggest that they could add the total number of articles they collected.

Response:

Figure 1. Excellent figure, however, the size is not adequate for its reading or visualization, I suggest that it be enlarged to improve its visualization.

Line 164. Type error.

Response:

Line 232. good description of results that provide a comprehensive view of this topic focused on large species, however, if the authors agree I suggest that the view on the use of analgesics also be integrated, i.e., from a perspective in small species the most used drugs are non-steroidal analgesics, but is this similar in large species? I suggest that the authors consider such idea within their study that could improve the impact of the proposal of their manuscript.

Response:

Line 248. delete (.

Response:

Line 279. Please replace "E. coli" with "E. coli".

Response:

Line 277. In supplementation to my comment on line 89, I suggest modifying the formatting of quoted within the text in the phrase "According to [14]".

Response:

Line 296. Again, please, to give a continuity to the reading of your manuscript I suggest that the author's last name should be written first followed by the citation number, please I invite authors to correct this in the rest of your manuscript.

Response:

Line 297 Please these acronyms are not previously defined, I invite to define them.

Response:

Lines 460 - 470. The first paragraphs of their discussion provide a description of the methodology of their study that was previously described. I suggest to the authors that they might mention here the most relevant finding of their study such as the influence of the effect of sex or age on the assessment.

Response:

Line 563. This idea is relevant to their study, if the authors allow me I suggest they include this sentence as the level of empathy may be influenced by the species of the animal the observer evaluates. Please refer to this article Bradshaw and Paul (2023) Could empathy for animals have been an adaptation in the evolution of Homo sapiens? Doi: https://doi.org/10.1017/S096272860000230X in order to supplement your references.

Response:

Line 625 In addition to mentioning limitations, they could mention possible future study trends.

Response:

Reviewer 3 Report

Comments and Suggestions for Authors

       The authors had a systematic review and meta-analysis focusing on numerical rating scale and visual analogue scale for pain assessment in cattle. These pain assessment criteria and their modifications are used to assess pain in animals.  This review article highlights the various factors affecting the outcome of assessment, which might provide an avenue to overcome the several influencing variables on pain assessment in cattle.     

Comments on the Quality of English Language

English editing needs to be improved. 

Reviewer 4 Report

Comments and Suggestions for Authors

Dear authors,

I read your manuscript, it is interesting and well-constructed. Notwithstanding this, I suggested some revisions in the attached pdf.

Regards

Comments on the Quality of English Language

The English Language is overall good.

Thanks

Round 2

Reviewer 2 Report

Comments and Suggestions for Authors

General comments

I appreciate the authors' consideration of my previous observations on a manuscript that I have enjoyed reading because of the contribution it makes to the area of pain management. I therefore consider it suitable for publication, although I believe that only a few simple changes should be made beforehand. 

Particular comments

Lines 29-30. Please if the authors agree in order to make it understandable to the reader that the sentence "a search for articles in the Pubmed (MEDLINE) and agricultural databases was performed where a total of 842 articles were identified" should be modified.

Response:

Lines 168 - 169. I appreciate the authors for considering the authors my comments, however, I apologize for not being clear in my explanation, I mean that there is a space the continuation of the sentence that makes it difficult to read, please remove it.

Response:

Line 227 Please I suggest you review the citation format according to the journal it should be accompanied with the reference number not at the end of the sentence.

Response:

Line 244. In complement to my previous comment please revise the format of cited and in the rest of the document.

Response:
